# Cyanate Induces Oxidative Stress Injury and Abnormal Lipid Metabolism in Liver through Nrf2/HO-1

**DOI:** 10.3390/molecules24183231

**Published:** 2019-09-05

**Authors:** Ling Hu, Kuan Tian, Tao Zhang, Chun-Hua Fan, Peng Zhou, Di Zeng, Shuang Zhao, Li-Sha Li, Hendrea Shaniqua Smith, Jing Li, Jian-Hua Ran

**Affiliations:** 1Neuroscience Research Center, College of Basic Medicine, Chongqing Medical University, Chongqing 400016, China (L.H.) (K.T.) (T.Z.) (C.-H.F.) (H.S.S.); 2Lab of Stem Cell and Tissue Engineering, Department of Histology and Embryology, Chongqing Medical University, Chongqing 400016, China (P.Z.) (D.Z.) (S.Z.) (L.-S.L.) (J.L.)

**Keywords:** cyanate, HL-7702, Nrf2/ ho-1, ROS, lipid metabolism

## Abstract

Chronic kidney disease (CKD) is problem that has become one of the major issues affecting public health. Extensive clinical data suggests that the prevalence of hyperlipidemia in CKD patients is significantly higher than in the general population. Lipid metabolism disorders can damage the renal parenchyma and promote the occurrence of cardiovascular disease (CVD). Cyanate is a uremic toxin that has attracted widespread attention in recent years. Usually, 0.8% of the molar concentration of urea is converted into cyanate, while myeloperoxidase (MPO) catalyzes the oxidation of thiocyanate to produce cyanate at the site of inflammation during smoking, inflammation, or exposure to environmental pollution. One of the important physiological functions of cyanate is protein carbonylation, a non-enzymatic post-translational protein modification. Carbamylation reactions on proteins are capable of irreversibly changing protein structure and function, resulting in pathologic molecular and cellular responses. In addition, recent studies have shown that cyanate can directly damage vascular tissue by producing large amounts of reactive oxygen species (ROS). Oxidative stress leads to the disorder of liver lipid metabolism, which is also an important mechanism leading to cirrhosis and liver fibrosis. However, the influence of cyanate on liver has remained unclear. In this research, we explored the effects of cyanate on the oxidative stress injury and abnormal lipid metabolism in mice and HL-7702 cells. In results, cyanate induced hyperlipidemia and oxidative stress by influencing the content of total cholesterol (TC), high-density lipoprotein (HDL), low-density lipoprotein (LDL), superoxide dismutase (SOD), catalase (CAT) in liver. Cyanate inhibited NF-E2-related factor 2 (Nrf2), heme oxygenase-1 (HO-1), and the phosphorylation of adenosine 5′monophosphate-activated protein kinase (AMPK), activated the mTOR pathway. Oxidative stress on the cells reduced significantly by treating with TBHQ, an antioxidant, which is also an activator of Nrf2. The activity of Nrf2 was rehabilitated and phosphorylation of mTOR decreased. In conclusion, cyanate could induce oxidative stress damage and lipid deposition by inhibiting Nrf2/HO-1 pathway, which was rescued by inhibitor of Nrf2.

## 1. Introduction

Chronic kidney disease (CKD), which results from a wide variety of disorders including diabetes, hypertension, and glomerulonephritis, is a worldwide medical and public health problem affecting about 16% of the worldwide population [1,2,3,4]. In recent years, a large amount of clinical data has confirmed that hyperlipidemia levels in the CKD population are significantly higher than in the general population, and characterized by elevated plasma triglyceride and low-density lipoprotein levels [5,6]. Dyslipidemia is not only a common complication of CKD patients, contributing to the progress of the disease, but also promotes the occurrence of cardiovascular disease (CVD), which is the major cause of death in patients with CKD [7,8,9]. At the same time, the serum lipid profile, as well as various aspects of HDL metabolism, were demonstrated to be profoundly altered in patients with nephrotic range proteinuria or CKD [10,11]. These abnormalities can, in turn, contribute to the progression of cardiovascular complications and various other comorbidities, such as foam cell formation, atherosclerosis, and/or glomerulosclerosis. Hypercholesterolemia and hypertriglyceridemia have been shown to be an independent risk factor for kidney disease progression [12,13]. These data demonstrate that the lipid metabolism disorder and kidney injury are mutually causal, and the pathological factors and mechanisms involved are extremely complicated. The previous research have not fully elucidated the relationship(s) between lipid metabolism disorder and kidney injury.

CKD is characterized by the progressive retention of metabolites normally excreted by the kidney, collectively termed “uremic toxins,” many of which have adverse effects on numerous organs. Cyanate is a uremic toxin that is converted from 0.8% of the urea. Probably due to the increased availability of urea, cyanate levels are also elevated in CKD [14,15]. In addition, recent studies have shown that myeloperoxidase (MPO) catalyzes the oxidation of thiocyanate to produce large amounts of cyanate at the site of inflammation during smoking, inflammation, or exposure to environmental pollution [16]. While CKD patients themselves are in a low-inflammation state, higher concentrations of cyanate are present in patients with CKD. The most significant biological activity of cyanate is related to the induction of carbamylation [17], which has been recognized as a post-translational modification of amino acids and protein modification that affects organs and body functions through a variety of mechanisms [16,18]. It has been reported that cyanate carbamoylation modified albumin, hemoglobin, low-density lipoprotein, high-density lipoprotein, etc., leading to atherosclerosis, endothelial dysfunction, and CVD [17,19]. In recent years, studies have also reported that cyanate can cause vascular endothelial cell dysfunction by inducing reactive oxygen species (ROS) production [20]. Oxidative stress causes disorder of liver lipid metabolism, which is an important mechanism leading to liver cirrhosis and liver fibrosis. Therefore, the mechanism of lipid metabolism disorder in CKD patients is related to the effects of carbamylated low-density lipoprotein and high-density lipoprotein. However, whether cyanate in patients with CKD that can directly cause liver tissue damage and abnormal blood lipids through oxidative stress lack evidence.

Oxidative stress and inflammation are important features of CKD. It is also an important reason driving the progression of CKD and development of cardiovascular diseases and other complications of CKD [21,22]. Oxidative stress is a condition in which the generation of ROS exceeds the capacity of the antioxidant defense system. It can occur as a result of increased ROS production, impaired antioxidant capacity, or both [23,24]. Increasing ROS generation or decreasing antioxidant availability lead to oxidative stress and induction of the proinflammatory response, which contribute to disease pathogenesis [22,25,26,27]. The natural antioxidant defense system consists of a large number of endogenous ROS scavenging molecules, antioxidant enzymes, and phase 2 detoxifying enzyme [28,29]. Nuclear factor-erythroid-2-related factor 2 (Nrf2) makes a central role in the basal activity and coordinates induction of over 250 genes, including those encoding antioxidant and phase 2 detoxifying enzymes and related proteins, such as catalase, SOD, heme oxygenase-1 (HO-1) [30,31], glutamate-cysteine ligase, glutathione S-transferase, glutathione peroxidase, and thioredoxin [32]. Normally, Nrf2 is held in an inactive state by bounding to Keap1 in the cytoplasm [33]. In response to oxidative stress signals from various sources, the Keap1/Nrf2 complex is disrupted [34], which promotes the accumulation of the de novo synthesized Nrf2 and its translocation to the nucleus [35,36,37]. Within the nucleus, Nrf2 binds to regulatory sequences, known as antioxidant response elements (ARE), in the promoter regions of genes encoding antioxidant and phase 2 detoxifying enzymes [38,39].

*tert*-Butyl hydroquinone (TBHQ) is known as an activator of Nrf2, which results in the induction of cellular antioxidant genes. In this research, we investigated the effects of cyanate on liver injure via the Nrf2, AMPK, and mTOR signals. It revealed that cyanate caused oxidative stress damage and lipid accumulation on liver both in vivo and in vitro, while TBHQ rescued oxidative stress damage and lipid accumulation on hepatocytes through the Nrf2, AMPK and mTOR signals.

## 2. Results

### 2.1. Cyanate Decreases Body Weight and Causes Dyslipidemia

Male C57/BL6 mice (6 weeks) weighing approximately 15–18 g were purchased from the Animal Experimental Center of Chongqing Medical University. Mice were fed a standard diet and maintained in individual ventilated cages (IVC) on conditions (temperature: 24 ± 1 °C, relative humidity: 40%–80%, *n* = 5 per cage). Seventy mice were randomly divided into two groups: 40 in the cyanate group and 30 in the normal control group. The cyanate group drank water containing 1% cyanate and the body weight of the mice was recorded every three days. After 30 days, the mice were sacrificed after being given anesthesia, and the blood biochemical indicators such as liver function, kidney function, and lipid were analyzed by the automatic biochemical analyzer. The tissues of the mice were collected for the following experiments.

The results (Figure 1) showed that the body weight of these mice decreased significantly compared with the control group (*p* < 0.01). The biochemical analysis results show that ALT, AST, ALP of mice in the cyanate group increased significantly compared with the control group (*p* < 0.05), the activity of LDH, CHE in the cyanate group decreased significantly compared with the control group (*p* < 0.05), the levels of TC and LDL content increased significantly compared with the control group (*p* < 0.05), the levels of HDL content decreased significantly compared with the control group (*p* < 0.05). As shown above, the cyanate significantly reduced liver function and caused dyslipidemia.

### 2.2. Cyanate Reduced the Antioxidant Capacity of Liver

After the mice were sacrificed, the liver was perfused with normal saline (0.9% NaCl, containing 0.16 mg/mL sodium heparin) to remove the blood samples, and tissue samples were obtained (*n* ≥ 30 per group). Take appropriate amount of tissue samples, add pre-cooled PBS and homogenize in an ice bath, then centrifuge the homogenate at 4 °C, take the supernatant as the sample to be tested, measure the protein concentration of the supernatant with BCA kit, according to the manufacturer’s instructions. The procedure was carried out to determine the activity of superoxide dismutase (SOD), catalase (CAT) and the contents of malondialdehyde (MDA) and nitric oxide (NO). Each experiment was repeated three times.

SOD is a free radical scavenger, which can scavenge superoxide anion free radicals in the body. Lower SOD content in the body implies a decreased ability to remove harmful free radicals. CAT is also an enzyme scavenger, known as catalase, a binding enzyme based on iron porphyrin. It promotes the decomposition of H_2_O_2_ into molecular oxygen and water, removes hydrogen peroxide from the body, and protects cells from H_2_O_2_ toxicity. It is one of the key enzymes in the biological defense system. MDA is a product of free radical action on lipid peroxidation in living organisms, causing cross-linking polymerization of macromolecules such as proteins and nucleic acids. And nitric oxide (NO) is a highly reactive free radical in the biography. It has a wide range of physiological effects in the body, such as relaxing vascular smooth muscle or mediating cytotoxic effects.

As shown in Figure 2, cyanate reduced the activity of SOD and CAT and increased the content of MDA and NO significantly, which suggested that cyanate could reduce the body’s antioxidant capacity.

### 2.3. Cyanate Causes Liver Injury and Lipid Accumulation

The pathological injuries were examined to detect the effect of cyanate on the liver. As shown in Figure 3A, the control group presented normal liver lobule structures, neatly arranged, with no hemorrhaging, no hepatocyte degeneration, necrosis, no fat deposition or inflammatory cell infiltration. In contrast to the control group, the cyanate-treated group displayed disordered hepatic lobule, hepatocyte steatosis, and balloon-like changes.

The results of PAS staining showed that the glycogen content of PAS in the cyanate group was higher than that in the control group (Figure 3B). The results of MASSON staining showed that the liver fiber changes in the cyanate group were not obvious (Figure 3C). Oil Red O staining (Figure 3D) showed that in the cyanate group, the nucleus (blue) had been surrounded by lipid droplets (red). Due to the increase of lipid droplets, the lipid metabolism may have been disturbed, and the related protein were detected. As shown in Figure 3E, the level of HMGCR increased while the level of LDLR decreased in the liver of cyanate-treated mice measured by western blotting (*p* < 0.01). Overall, these results indicate that cyanate increased ROS levels and caused lipid deposition.

### 2.4. Cyanate Inhabits the Nrf2 Pathway in Liver

As described above, when stimulated by external factors, Nrf2 can be activated and translocated into the nucleus to initiate its transcriptional activity. We examined the effect of cyanate on Nrf2 activity and pathway molecules by western blotting. As results showed in Figure 4,B, the level of Nrf2 and HO-1 decreased and there were no significant changes of Keap1 in the liver of cyanate-treated mice. In addition, the results demonstrate that the level of p-AMPK decreased and the level of p-mTOR, p-S6K, p-S6 relatively increased in cyanate group (Figure 4C,D).

### 2.5. Cyanate Decreases ROS Levels Caused Lipid Deposition in HL-7702 Cells

Cyanate the activity of antioxidant, causes liver damage and reduces the expression of the Nrf2 signaling pathway. We explored the potential mechanism of how cyanate causes liver damage in vitro. Initially, we detected the anti-proliferation effect of cyanate on HL-7702 cells. Cell viability was investigated by the CCK-8 assay after HL-7702 cells were treated with different concentrations (0, 0.5, 1.0, and 2.0 mM) of cyanate for 24 h. As shown in Figure 5, the viability of HL-7702 cells was significantly reduced when treated with cyanate for 24 h. Moreover, cell viability decreased significantly when cells were treated with 2 mM cyanate (Figure 5A). Thus, we used cyanate at a concentration of 1 mM for subsequent experiments. To further verify that cyanate-induced oxidative stress was mediated by the Nrf2-dependent pathway, we used the TBHQ (10 μM), which is the major metabolite of butylated hydroxyanisole, the key factor in the reaction of cellular oxidative stress. Then we measured the lipid droplets of the cells by oil red O staining, the level of intracellular reactive oxygen species by DCFH-DA method, and observed the ultrastructure of the cells by the electron microscope. The results showed HL-7702 cells treatment with cyanate or TBHQ for 24 h had no effect on cell morphology (Figure 5B).

After treatment with cyanate for 24 h, intracellular lipid droplets increased and production of ROS was increased, but these changes were reversed after pre-treatment with TBHQ (Figure 5C,D). The results of electron microscopy showed that the cell nucleus was uniform, only a small amount of lipid droplets were observed in cytoplasm, and the mitochondria structure was normal. The nucleus of the cyanate-treated group was pyknotic, and a large number of round lipid droplets of different sizes were observed in the cytoplasm, the mitochondria are swollen and deformed (Figure 5E).

### 2.6. TBHQ Alleviates Oxidative Stress Caused by Cyanate in HL-7702 Cells

We also detected CAT, SOD, MDA, and NO levels in different groups. The results showed that the activities of CAT and SOD in the cells were significantly decreased and the content of MDA and NO in the cells were significantly increased after treatment with 1 mM cyanate for 24 h. After pretreatment with TBHQ before adding cyanate, the activities of SOD, CAT and the content of MDA and NO in the cells returned to normal levels (Figure 6). These results suggest that TBHQ can alleviate oxidative stress caused by cyanate.

### 2.7. TBHQ Rescues the Inhibition Activity of the Nrf2 Pathway and the Activation of mTOR Pathway Caused by Cyanate

To further validate that cyanate-induced oxidative stress was mediated by the Nrf2-dependent pathway, we used the specific agonist TBHQ (10 μM), which is the major metabolite of butylated hydroxyanisole, and induces an antioxidant response through the redox-sensitive transcription factor, nuclear factor-E2-related factor-2 (Nrf2), to treat HL-7702 cells for 2 h before exposure to 1 mM cyanate. The levels of oxidative stress-related proteins Nrf2 pathway were measured by western blotting methods. We found TBHQ can rescue the decreased levels of Nrf2, HO-1 p-AMPK and the increased activities of mTOR pathway in HL-7702 cell treated with cyanate (Figure 7A–D). Subsequently, we further investigated the expression of Nrf2 and HO-1 in HL-7702 cells treated with 1 mM cyanate for 24 h by the immunofluorescence method. After treatment with cyanate, the levels of Nrf2, HO-1 in the cells were significantly reduced compared to control (untreated) cells (Figure 7E–G). However, in the case of TBHQ pretreatment, the reduction levels of Nrf2 and HO-1 caused by cyanate were remarkably restored. Taken together, these results suggest that TBHQ can rescue the inhibition activity of the Nrf2 pathway and the activation of mTOR pathway caused by cyanate.

## 3. Discussion

With the improvement of the population’s economic conditions and changes in living habits, chronic kidney disease has become one of the major problems affecting public health [4,40], and is the third top killer after cancer and heart disease. Chronic kidney disease (CKD) is associated with the development of atherosclerosis and premature death from cardiovascular disease [41,42]. The predisposition of patients with CKD to atherosclerosis is driven by inflammation, oxidative stress and dyslipidemia, all of which are common features of this condition [43,44]. Dyslipidemia is associated with atherosclerotic vascular changes and the risk of occurrence of acute myocardial infarction in hemodialysis patients. CKD causes profound dysregulation of lipoprotein metabolism, resulting in multiple lipoprotein abnormalities. During the early stages of CKD, significant changes in apolipoproteins usually precede changes in plasma lipid levels. High-density lipoprotein (HDL) levels and increased triglyceride-rich lipoproteins are the major lipid abnormalities. Dyslipidemia is a common clinical manifestation of many primary or secondary kidney diseases and is itself involved in the development and the evolution of kidney disease [5,45]. Clinical studies suggest that dyslipidemia can aggravate kidney tissue damage, and hypercholesterolemia and hyper-triglyceridemia have been shown to be independent risk factors for progression of kidney disease [46,47,48]. Abnormal lipid metabolism and renal damage in chronic kidney disease have attracted widespread attention [49,50,51,52]. Therefore, investigating the blood lipid levels in patients with chronic kidney disease is of great significance to the study of its therapeutic strategies [13]. After drinking 1% cyanate-containing water, the body weights of mice were significantly decreased compared to those of the normal group, which has been reported [53]. However, in their study, mice were treated with cyanate (1 mg/mL in drinking water), while we used 10 mg/mL cyanate in drinking water to feed the mice because we believe that CKD patients may accumulate high local concentrations of cyanate due to urea accumulation and inflammatory conditions in the body. Perhaps because of the different concentrations of cyanate used, those authors thought that cyanate had a protective effect on the liver, while our experimental results show that cyanate has a detrimental effect on the liver. The results of serum biochemical analysis of mice in the cyanate group showed that ALT, AST, ALP increased, and LDH, CHE decreased significantly compared with the control group, suggesting that liver damage and metabolic function decreased in mice. While the levels of TC and LDL content increased significantly and the levels of HDL content decreased significantly compared with the control group. According to these findings, we can infer that cyanate decreases body weight and causes dyslipidemia.

The main physiological function of the liver is its involvement in glucose and lipid metabolism, and it is also a key organ regulating nuclear transcription factors that affect lipid metabolism, insulin sensitivity and inflammatory responses, and oxidative stress [54]. The results of Sokołowska et al. also show that cyanate can disturb the metabolism and functioning of the liver [55]. In our study, hepatic lobular structural disorders, hepatic steatosis and lipid deposition in mice in the cyanate group were observed by liver pathology examination.

CKD patients have defects in the antioxidant defense system [56]. Increased levels of several plasmatic oxidative markers in CKD have been demonstrated [57,58,59]. Moreover, several clinical studies suggest that antioxidant therapy in CKD patients may prevent the progression to end-stage renal disease and reduce CV mortality [29,60]. Under normal conditions, ROS produced during metabolism are contained by the natural antioxidant defense system. The body forms a complex oxidative stress response system when exposed to electrophilic agents or reactive oxygen species, induces a range of protective proteins to alleviate damage to cells [61,62]. However, when ROS production exceeds the capacity of this system, it leads to oxidative stress in which the uncontained or uncontainable ROS cause tissue damage and dysfunction by attacking, denaturing, and modifying structural and functional molecules and by activating redox-sensitive transcription factors and signal transduction pathways. These events result in necrosis, apoptosis, inflammation, fibrosis, and other disorders that participate in the disease process. Thus, oxidative stress occurs as a result of increased ROS production and/or an impaired antioxidant defense system [54,63]. As an antioxidant system in the body, SOD is a free radical scavenger, which can scavenge superoxide anion free radicals. Lower SOD content indicates a reduction of the ability to remove harmful free radicals. CAT, also known as catalase, is a binding enzyme based on iron porphyrin that acts as an enzyme scavenger. It promotes the decomposition of H_2_O_2_ into molecular oxygen and water, removes hydrogen peroxide from the body, and protects cells from H_2_O_2_ toxicity. It is one of the key enzymes in the biological defense system. MDA is the end product of free radical action on lipid peroxidation in living organisms, causing cross-linking polymerization of macromolecules such as proteins and nucleic acids, and is cytotoxic. NO (vascular endothelium-lowering factor) is a highly reactive free radical in the biography. SOD and CAT activity is an important index to measure the ability of anti-oxidation and scavenge free radical. MDA and NO are products of oxidative stress in the body. In this study, the SOD, CAT content in the cyanate group decreased significantly compared with the control group, and the content of MDA and NO increased significantly, which indicated an impaired antioxidant capacity and acceleration of the hyperlipidemia process. However, it was found that the content of SOD, CAT increased significantly after the application of TBHQ, MDA and NO content decreased significantly after administration of TBHQ, implying the gradual recovery of the body’s anti-oxidative stress ability. In addition, we examined the production of ROS in cyanate-treated cells by DCFH-DA in vitro. Cyanate can significantly increase the level of ROS in HL-7702 cells, but pre-treatment with TBHQ before loading cyanate reduced ROS levels, indicating that TBHQ can effectively protect cells from cyanate-induced oxidative stress damage.

Oxidative stress and inflammation are involved in the development and progression of CKD and its complications, with these two processes being inseparably linked as each provokes and amplifies the other. NF-E2-related factor 2 (Nrf2) is a key factor in cellular oxidative stress [64,65,66]. It regulates the expression of the antioxidant enzyme and a series of detoxification enzymes by interacting with antioxidant response element ARE [31]. Cells are subjected to numerous endogenous and exogenous stresses, and there is increasing evidence that Nrf2 has a cytoprotective effect. The Kelch-like ECH-associated protein 1 (Keap1)—Nrf2 pathway is the major regulator of cytoprotective responses to oxidative and electrophilic stress [66,67]. Under physiological conditions, Keap1 binds to Nrf2, triggering its proteasomal degradation. In the presence of oxidative stress, Nrf2 escapes degradation and undergoes translocation to the nucleus, where it binds to AREs and up-regulates the expression of various downstream genes [68,69]. Activation of the Nrf2/ARE axis induces a strong antioxidant response, making pharmacological activation of this pathway a promising target for various diseases, including kidney disease [70,71,72]. In addition, under oxidative stress, Nrf2 is activated to induce the expression of antioxidant enzymes such as heme oxygenase-1 (HO-1) [73]. HO-1 is a rapidly inducible stress-responsive enzyme that degrades heme to carbon monoxide (CO), biliverdin and ferrous iron. Induction of HO-1 provides a protective function in the cardiovascular system through the effects of CO and bilirubin on various cells. HO-1 executes anti-apoptotic, anti-inflammatory, anti-hypertensive and anti-oxidant functions [64,74,75]. However, it was later found that some stimulating substances did not dissociate Keap1-Nrf2, and Nrf2 may be degraded by ubiquitination [76], the exact mechanism of how cyanate decrease Nrf2 expression remains to be determined in the future. In this study, we examined the expression of Nrf2 pathway protein in the liver of mice and HL-7702 cell. The levels of Nrf2 and HO-1 decreased in mice or cells treated with cyanate, and there was no significant change in the level of keap1. The results of cellular immunofluorescence tests showed that after cells were cyanate-loaded, Nrf2 levels in the nucleus was significantly reduced, indicating that cyanate may also ubiquitinate Nrf2 to cause degradation while stimulating the production of reactive oxygen species. The translocation of Nrf2 into the nucleus is reduced, the antioxidant capacity of the cells is further weakened, and the oxidative stress is aggravated.

Oxidative stress involves cellular or molecular damage caused by ROS, resulting from insufficient levels of antioxidants and/or antioxidant enzyme systems [77,78]. Excessive ROS can result in deoxyribonucleic acid damage, protein misfolding, organelle injury, and neuronal synaptic dysfunction [79]. AMPK, a key energy sensor of cellular metabolism, plays an important role in neurodegeneration, inflammation, and oxidative stress. There are reports in the literature, treatment with 100 μmol/kg NaHS (i.p. injection) every two days for 12 weeks could protect mouse arterial endothelial cells by suppressing excessive autophagy induced by oxidative stress through the Nrf2-ROS-AMPK signaling pathway [80]. Under normal conditions, the body activates AMPK phosphorylation under oxidative stimulation to regulate energy metabolism and oxidative stress balance, but when the oxidative stress system is imbalanced in the body, excessive ROS feedback regulation inhibits AMPK phosphorylation [81,82]. As mTOR, comprising two functional complexes (mTORC1 and mTORC2), is a serine/threonine-protein kinase which regulates intracellular signaling related to cell growth, cell survival and protein synthesis [83,84,85]. mTORC1 acts positively as a nutrient/energy/growth hormone-sensitive regulator. Dysregulation of the mTOR signaling pathway has been associated with several major disease states (e.g., diabetes, obesity, cancer, and cognitive defects). On the other hand, AMPK plays an important role in cellular energy homeostasis under conditions of low energy substrate and intracellular ATP [86]. High glucose, amino acids or lipids in blood can inhibit intracellular AMPK activity, possibly related to metabolic diseases [87,88]. It was recently reported that AMPK inhibits the activity of mTORC1 via the phosphorylation of raptor or tuberous sclerosis complex (TSC) 2 [89,90]. Thus, the mTOR signaling pathway is inversely modulated by the activity of AMPK. The mammalian target of rapamycin (mTOR), a kinase that is activated by anabolic signals, plays fundamental roles in regulating lipid biosynthesis and metabolism in response to nutrition. In this research, we examined the effects of cyanate on the expression of AMPK and mTOR in vivo and in vitro. The results indicated that phosphorylation level of AMPK was significantly decreased and the level of phosphorylated mTOR-S6K-S6 was increased with cyanate treatment, which indicates that cyanate may affect energy and lipids metabolism in liver. In vitro experiments further confirmed the molecular mechanism by preincubation with TBHQ, a pro-oxidant, DMNQ generates ROS in cells and actually activates Nrf2, leading to enhanced antioxidant defense. The ROS and lipid droplets produced in the cells decreased, the phosphorylated AMPK level recovered, and the mTOR pathway level decreased to the normal level, indicating that cyanate may cause liver oxidative stress damage and lipid deposition by inhibiting the Nrf2 -AMPK pathway and activating the mTOR pathway.

In summary, as shown in Figure 8, the results suggest cyanate promotes oxidative stress injury and lipid aggregation in mice and HL-7702 cells, possibly by inhibiting Nrf2, AMPK and activating the mTOR pathway. Which may associated with oxidative stress and inflammation in patients with CKD complicated with hyperlipidemia. The result shown that TBHQ can rescue oxidative stress damage and lipid accumulation caused by cyanate. From the above, TBHQ may be a potential therapeutic drug for targeting hyperlipidemia in CKD patients.

## 4. Materials and Methods

### 4.1. Reagents and Antibodies

Cyanate was purchased from Sigma-Aldrich (St Louis, MO, USA). The SOD, CAT, MDA and NO kits were obtained from Nanjing Jiancheng Bioengineering Institute (Nanjing, China). The antibodies against HMGCR (ab174830), LDL Receptor (ab52818), HO-1(ab68477) were purchased from Nrf2am (Cambridge, UK). Antibodies against mTOR(7c10), p-mTOR(D9C2), S6K(49D7), p-S6K (108D2), S6(5G10), p-S6(ser-235), Nrf2 (D1Z9C), KEAP1 (D6B12), and β-actin (8H10D10) were purchased from Cell Signaling Technology (Danvers, MA, USA). Cell Counting Kit-8 was purchased from Dojindo (Kumamoto, Japan). The Oil red O and Reactive Oxygen Species Assay Kits were purchased from Beyotime Institute of Biotechnology (Shanghai, China).

### 4.2. Cell Culture and Culture Condition

Normal hepatocyte HL-7702 cells were a gift from the Gene Research Division of the Third Military Medical University Institute (Chongqing, China). Cells were cultured with Dulbecco’s Modified Eagle Medium (DMEM, Gibco-Invitrogen Corporation, Carlsbad, CA, USA) and 10% fetal bovine serum (FBS, Gibco-Invitrogen Corporation) and 1% antibiotic (Beyotime Institute of Biotechnology) in a humidified atmosphere with 5% CO_2_ at 37 °C.

### 4.3. Cell Viability Assay

Cell viability was determined using a Cell Counting Kit-8 (CCK-8). Briefly, following the manufacturer’s instructions, cells were plated in 96-well plates at a density of 5 × 10^3^ cells per well. After attachment for 24 h, the medium was changed for fresh medium containing Cyanate with indicated or different concentrations respectively and cells were incubated for another 24 h. After indicated cultivation time, the viability of cells was measured using the CCK-8 assay. Before testing, CCK-8 solution (10 µL) was added to each well containing a 100 µL mixture of culture medium. The plates were incubated for 2 h at 37 °C in the incubator. Cell viability was counted by absorbance measurements at 450 nm using an auto microplate reader (Bio-Rad, Hercules, CA, USA). The OD_450_ value was proportional to the viability of the cells. All experiments were performed in triplicate.

### 4.4. Reactive Oxygen Species Assay Kits

2′,7′-Dichlorofluorescin diacetate (DCFH-DA) is a non-labeling oxidation-sensitive fluorescent probe. DCFH-DA itself has no fluorescence and is free radical. After passing through the cell membrane and entering the cell, it can be hydrolyzed by intracellular esterase to form DCFH. DCFH does not penetrate the cell membrane, making it easy for the probe to be loaded into the cell. Reactive oxygen species in the cells can oxidize non-fluorescent DCFH to produce fluorescent DCFH. The fluorescence of the DCFH is detected to determine the level of intracellular reactive oxygen species.

### 4.5. Immunoblot Assay

Cells were isolated and washed with cold PBS and suspended again in lysis buffer containing 1% phenylmethanesulfonyl fluoride and phosphatase inhibitor added just before use (Beyotime Institute of Biotechnology). After incubation for 30 min on ice, the supernatant was collected by centrifugation at 12,000 rpm for 15 min at 4 °C and the protein concentration was determined by BCA Protein Assay Kit (Beyotime Institute of Biotechnology). Equal amounts of proteins from each sample were separated by sodium dodecyl sulfate-polyacrylamide gel electrophoresis (SDS-PAGE) and separated proteins were transferred to polyvinylidene difluoride (PVDF) membranes. The membranes were blocked with 5% non-fat milk for 2 h, incubated with primary antibody overnight at 4 °C, washed in Tris-buffered saline with Tween20 (TBST) for 30 min, and incubated with IgG horseradish peroxidase (HRP)-conjugated secondary antibody for 1h at room temperature. Bound immune-complexes were detected by enhanced chemiluminescence (ECL) Western blotting detection reagents (Millipore, Billerica, MA, USA) and exposure to the Luminescent Image Analyzer (Bio-Rad) [53].

### 4.6. Immunofluorescence Staining

Cells (3 × 10^4^)owere seeded in six-well plates, put in a glass slide for 24 h, and treated or not with cyanate for 24 h. Cells were then fixed in 4% paraformaldehyde for 30 min at room temperature and washed twice for 10 min followed by permeabilization with 0.5% Triton X-100 in PBS for 20 min. Fixed cells were blocked with 1% goat serum albumin for 1 h, then incubated with primary antibody against Nrf2, KEAP1, HO-1 at 4 °C overnight. After being washed thrice with PBS, cells were stained with FITC-conjugated secondary antibodies. (Cell Signaling Technology) at 37 °C for 2 h. Nuclei were stained with 4′,6-diamidino-2-phenylindole (DAPI) for the last 5 min. Images were captured by fluorescence microscopy (Olympus, Tokyo, Japan) [54].

### 4.7. Detection of Oxidation-Associated Biological Indicators

After the mice were sacrificed, the liver was perfused with normal saline to remove the blood and tissue samples were obtained (*n* ≥ 30 per group). Take an appropriate amount of tissue samples or collected cells, add to the pre-cooled PBS and homogenize in an ice bath, then centrifuge the homogenate at 4 °C, take the supernatant as the sample to be tested, and measure the protein concentration of the supernatant with the BCA kit. After that, according to the procedures of the manual, the activity of SOD, CAT and the contents of MDA and NO were measured. Each experiment was repeated three times.

### 4.8. Biochemical Analyses

Alanine aminotransferase (ALT), aspartate aminotransferase (AST), alkaline phosphatase (ALP), lactate dehydrogenase (LDH), cholinesterase (CHE), total cholesterol (TC), triglycerides (TG), low-density lipoprotein-cholesterol (LDL) and high-density lipoprotein-cholesterol (HDL) were determined by standard laboratory procedures.

### 4.9. Histological Analysis of Liver Steatosis

Hepatic steatosis was assessed by Oil Red O staining of OCT-embedded cryosections. Briefly, liver sections were fixed in 60% isopropanol for 10 min and stained with 60% Oil Red O for 30 min and subsequently washed with 60% isopropanol. Sections were counterstained with Gill’s hematoxylin, washed with 4% acetic acid solution and mounted with an aqueous solution. The sections were observed under the microscope.

### 4.10. Liver Hematoxylin and Eosin Staining

The left lobe of liver was collected after the mice were killed and put into a 4% formaldehyde buffer. The tissues were embedded in paraffin, sliced and stained with hematoxylin and eosin. The slices were observed with a light microscope.

### 4.11. Fibrosis Histology

Liver and WAT samples were fixed in 10% formalin, embedded in paraffin, cut into 5 μm sections and stained with hematoxylin-eosin (H&E) at the Pathology Department of the Hospital Clinic. Fibrosis was assessed by Masson’s trichrome staining at the Pathology Department of the Hospital Clinic. Sections were visualized at 100× magnification under a microscope.

### 4.12. Animal Models in C57BL/6 Mice

All procedures involving mice and experimental protocols were approved by the Laboratory Animal Center of Chongqing Medical University. C57BL/6 wild-type (WT) mice, male, 6-weeks-old, were purchased from the Laboratory Animal Center of Chongqing Medical University. Mice were fed with 1% strength cyanate solution and body weights were weighed every three days, and the mice body weight curves were plotted. At the end of the experiment, all animals were sacrificed by dislocation after anesthesia and then the blood and liver of all groups were collected for the following experiments. Each experiment was repeated three times.

### 4.13. Statistical Analysis

Figures in the text are representative of at least three independent experiments. The data are expressed as means ± SD (standard deviation). Significance of the differences between various experimental and control groups was statistically analyzed using Student’s t-test with the SPSS 22.0 software (IBM, Armonk, NY, USA) and indicated as * *p* < 0.05, ** *p* < 0.01.

## 5. Conclusions

The results of this study indicate that cyanate can reduce the expression of Nrf2/HO-1 pathway in the liver, causing liver oxidative stress damage and abnormal lipid metabolism. However, the application of TBHQ can restore the Nrf2 signaling pathway to normal and reduce oxidative stress. Abnormal damage and lipid metabolism suggest that this effect may be achieved by inhibiting the Nrf2 signaling pathway. These findings provide a possible target for the treatment of dyslipidemia. In addition, further experiments are needed to investigate the use of TBHQ as a drug to reduce the risk of liver damage and hyperlipidemia.

## Figures and Tables

**Figure 1 molecules-24-03231-f001:**
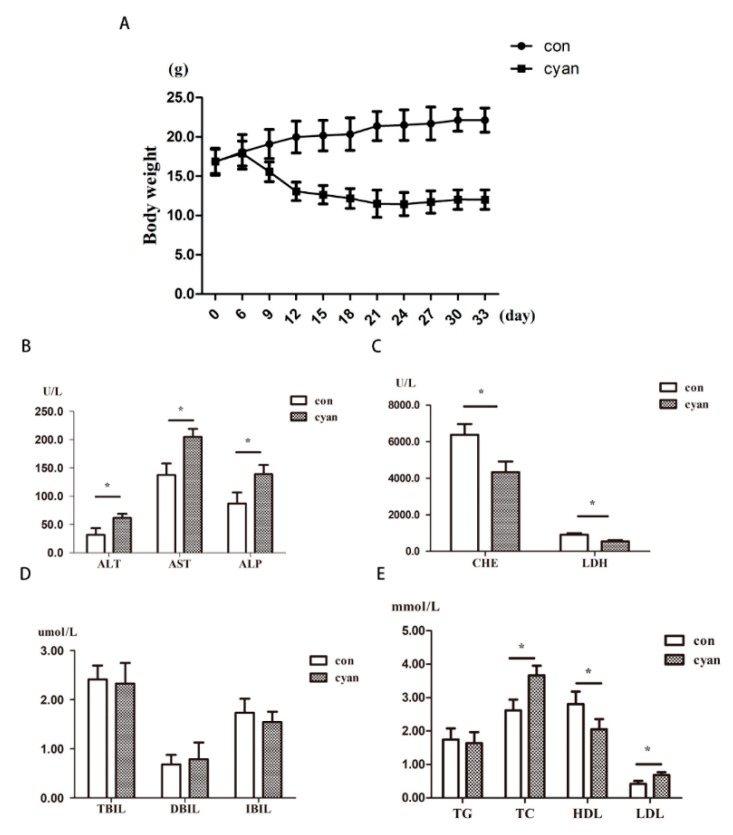
Cyanate decreased body weight and caused dyslipidemia. (**A**) Mice body weights decreased by drinking 1% cyanate water. (**B**)The levels of ALT, AST, ALP increased in cyanate group (*p* < 0.05). (**C**) The levels of CHE and LDH decreased in cyanate group. The levels of total cholesterol, high-density lipoprotein increased in cyanate group (*p* < 0.05). (**D**) The changes of TBIL, DBIL, IBIL were not statistically significant. (**E**) The level of HDL had decreased, and the levels of TC, LDL were increased in cyanate group (*p* < 0.05).

**Figure 2 molecules-24-03231-f002:**
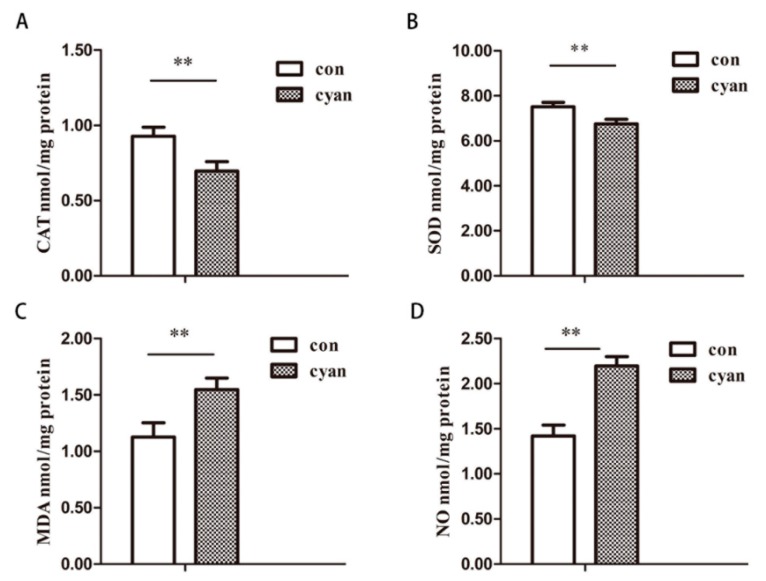
Cyanate reduced the antioxidant capacity of the liver. (**A**) SOD activity decreased significantly after cyanate stimulation (*p* < 0.01). (**B**) The activity of CAT decreased after cyanate treatment (*p* < 0.01). (**C**) The activity of MDA increased significantly after cyanate stimulation (*p* < 0.01). (**D**) After cyanate treatment, the content of NO increased (*p* < 0.01). *n* ≥ 8 per group, and the experiment was repeated three times. Values are means and standard errors (* *p* < 0.05 and ** *p* < 0.01 versus control).

**Figure 3 molecules-24-03231-f003:**
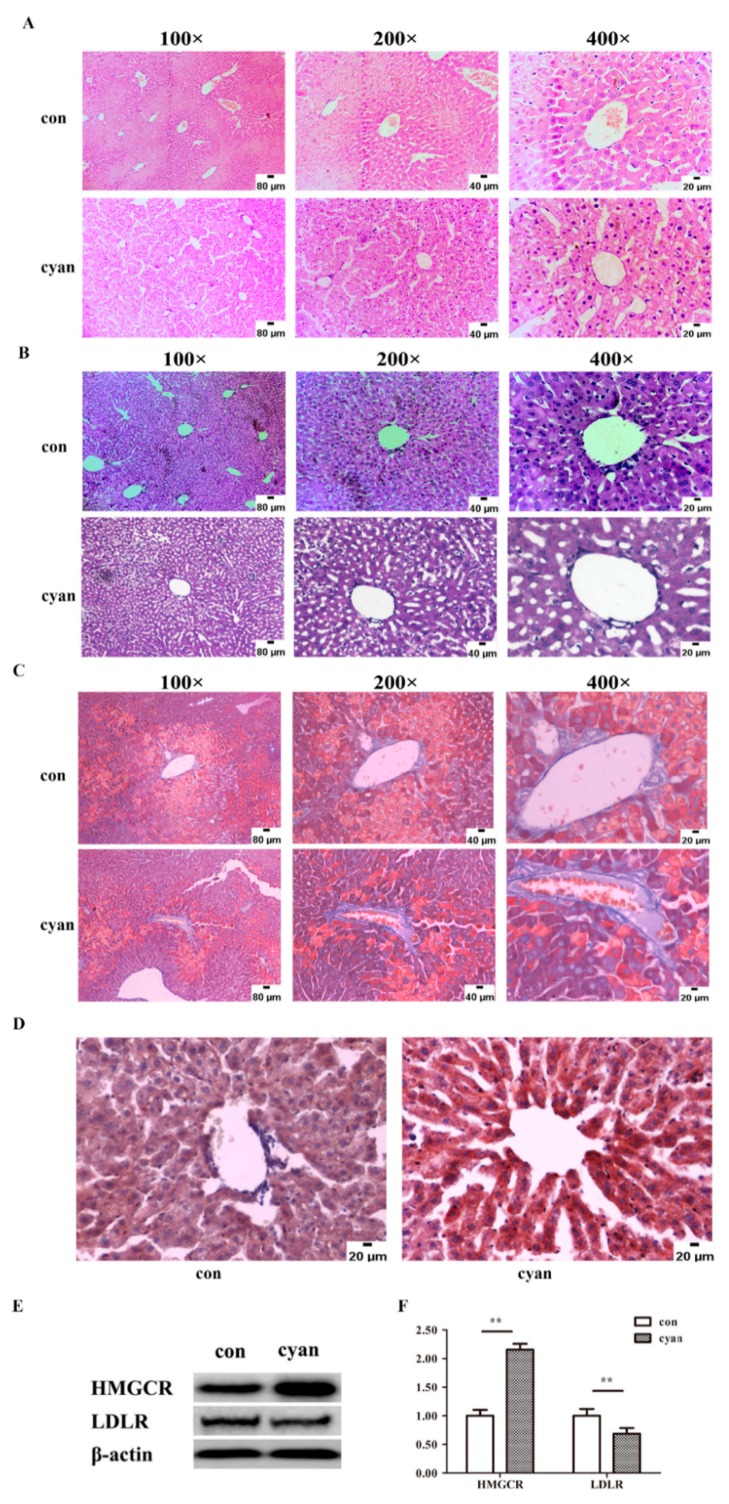
Cyanate caused liver injury and lipid accumulation. (**A**) Hematoxylin and eosin staining of liver tissues. (**B**) PAS staining of liver tissues. (**C**) MASSON staining of liver tissues. (**D**) Oil Red O staining of liver tissues. (**E**,**F**) Western blotting showed the levels of HMGCR and LDLR proteins in the liver of mice treated with or without cyanate (*p* < 0.01). *n* ≥ 8 per group, and the experiment was repeated three times. Values are means and standard errors (* *p* < 0.05 and ** *p* < 0.01 versus control).

**Figure 4 molecules-24-03231-f004:**
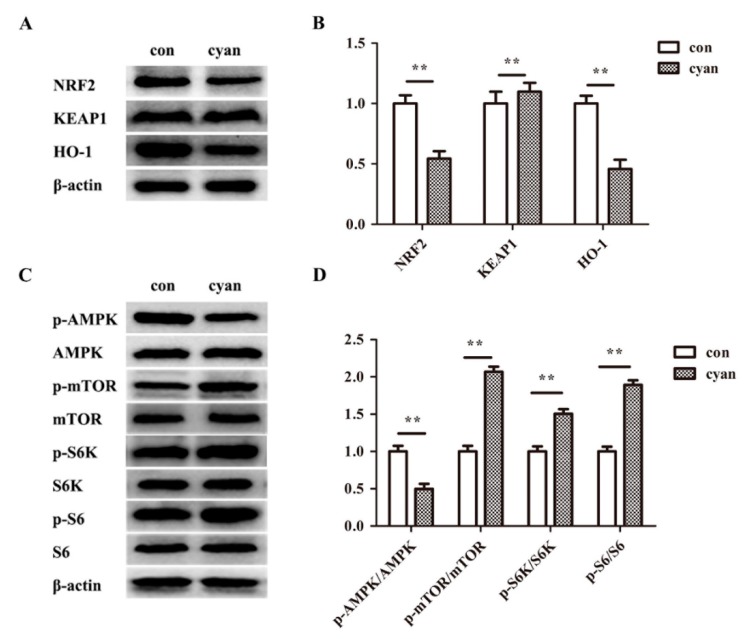
Cyanate decreased the expression of Nrf2 in the Liver. (**A**,**B**) The levels of Nrf2 and HO-1 in the liver were significantly decreased in the cyanate group. (**C**,**D**) The levels of p-AMPK decreased and p-mTOR, p-S6K, and p-S6 increased in liver after mice treated with cyanate. *n* ≥ 8 per group, and the experiment was repeated three times. Values are means and standard errors (* *p* < 0.05 and ** *p* < 0.01 versus control).

**Figure 5 molecules-24-03231-f005:**
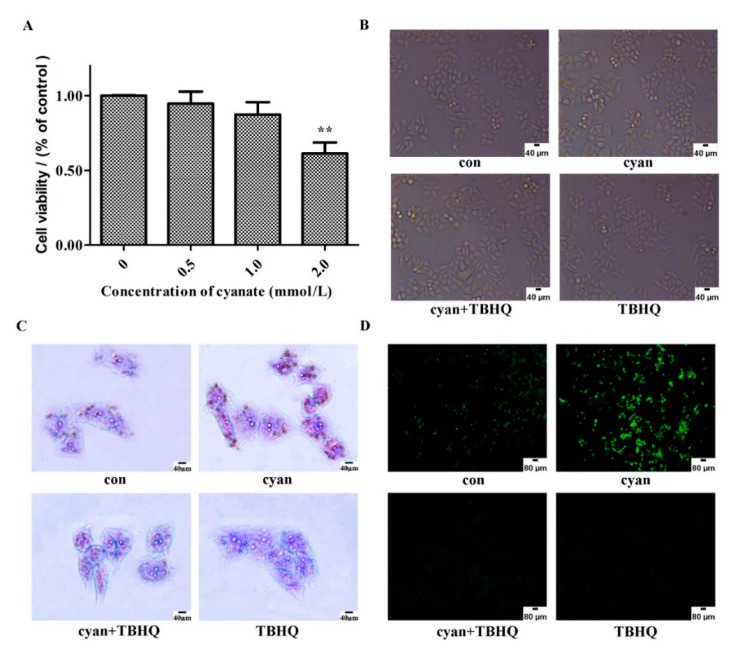
Cyanate decreased ROS levels and lipid deposition in HL-7702 cell. (**A**) HL-7702 cells were incubated with increasing concentrations of cyanate (0, 0.5, 1, and 2 µM) for 24 h. Cell Counting Kit-8 (CCK-8) assay was performed to detect the cytotoxic effect of cyanate. (**B**) Morphological changes of cells under inverted microscope. (**C**) Oil Red O staining of HL-7702 cells after being treated with cyanate for 24 h. The results showed that the cell lipid droplets increased significantly after 24 h of cyanate treatment, and the lipid droplets of the cells treated with cyanate were reduced after pre-treatment with TBHQ for 2 h before exposure to cyanate. (**D**)The results of DCFH-DA assay showed that the intracellular ROS increased significantly after 24 h of cyanate treatment, while there were normal ROS level in the cyanate-treated cells by TBHQ pre-treatment for 2 h before exposure to cyanate. (**E**) The ultrastructure of the cells was observed under an electron microscope.

**Figure 6 molecules-24-03231-f006:**
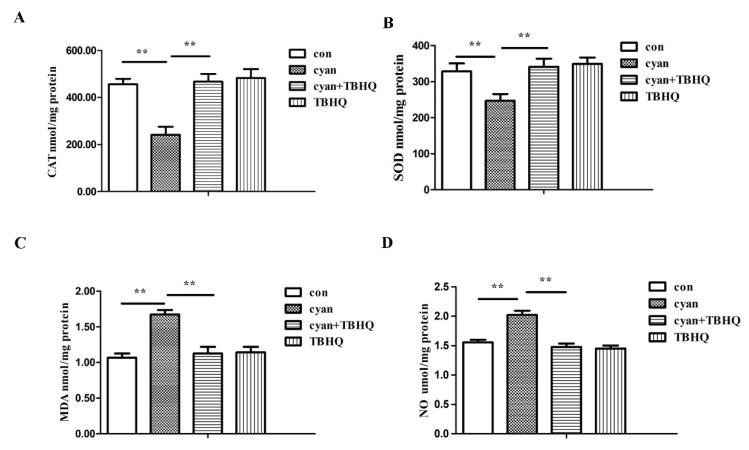
TBHQ alleviates oxidative stress caused by cyanate in HL-7702 cells. (**A**,**B**) TBHQ can rescue decreased levels of CAT and SOD in HL-7702 cells caused by cyanate. (**C**,**D**) TBHQ can rescue increased MDA and NO content in HL-7702 cells caused by cyanate. Each group was a mixture of cells collected three times (*n* = 3) and the experiment was repeated three times. Values are means and standard errors (* *p* < 0.05 and ** *p* < 0.01 versus control).

**Figure 7 molecules-24-03231-f007:**
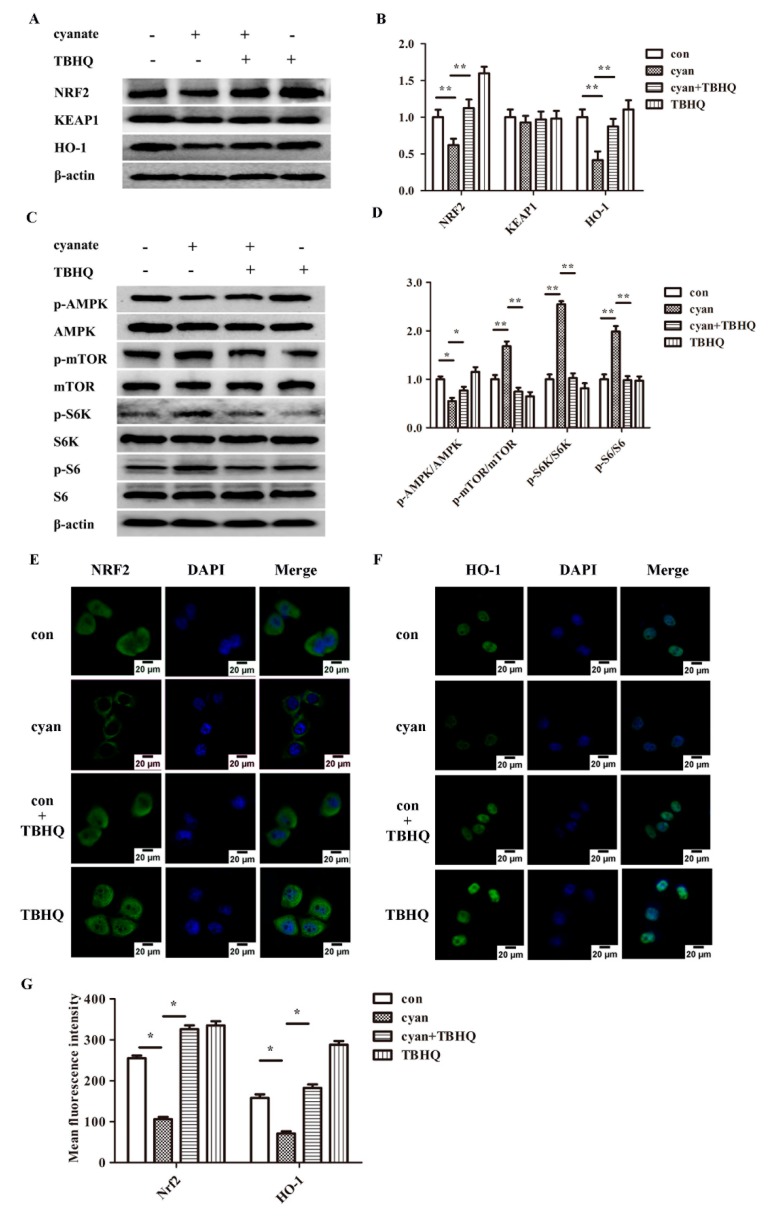
TBHQ rescue the inhibition of the activity of the Nrf2 pathway and the activation of the mTOR pathway caused by cyanate. (**A**,**B**) The western blotting results showed that the expression levels of Nrf2, Keap1, and HO-1 in cytoplasmic protein. (**C**,**D**) The results of western blotting showed the relative expression of AMPK and mTOR pathway in cytoplasmic protein. (**E**,**F**) The results of immunofluorescence showed that TBHQ regulates the inhibition of the activity of the Nrf2 and HO-1 caused by cyanate in HL-7702 cell. (**G**) Mean fluorescence intensity of Nrf2 and HO-1 was quantified and is presented as means ± SD. Each group was a mixture of cells collected three times (*n* = 3) and the experiment was repeated three times. Values are means and standard errors (* *p* < 0.05 and ** *p* < 0.01 versus control).

**Figure 8 molecules-24-03231-f008:**
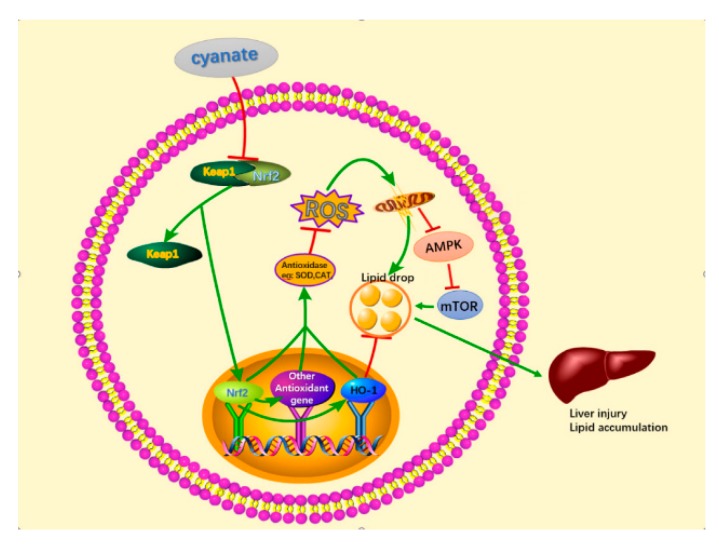
Model of Nrf2 signaling cascade. Cyanate may reduce the release of Nrf2 into the nucleus by inhibiting the uncoupling of the Nrf2-Keap1 complex, thus the expression of antioxidant enzymes (including HO-1) and a series of detoxifying substances are reduced, resulting in excessive release of ROS. Further impairs mitochondrial damage and abnormal energy metabolism. This results in a decrease in the level of p-AMPK and activation of the mTOR pathway, causing lipid deposition to further damage the liver. More importantly, these signaling pathways play a key role in oxidative stress and inhibition of lipid accumulation.

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
