# Peer review of "Cyanate Induces Oxidative Stress Injury and Abnormal Lipid Metabolism in Liver through Nrf2/HO-1"

_molecules, 2019, doi:10.3390/molecules24183231_

Round 1
Reviewer 1 Report
In this manuscript Hu and co-authors investigated the effects of cyanate on liver function in a mouse model and oxidative stress in a cell culture model. The authors found that administration of toxic dose of cyanate to mice causes loss of body weight and reduced expression of liver enzymes.
In a cell culture model the authors report that cyanate caused reduction in Nrf2 and HO1 expression, reduction in pAMPK levels and increase in the levels of pmTOR, pS6K and pS6. Moreover, it causes increased ROS production and lipid droplet accumulation which could be reversed with an antioxidant TBHQ. The authors then conclude
"that cyanate can cause oxidative stress and abnormal lipid metabolism in the liver, and in addition, this effect is achieved by inhibiting the Nrf2 signaling pathway. These findings provide an experimental basis for the treatment of dyslipidemia in patients with CKD. Further experiments are needed to investigate the use of TBHQ as a drug to reduce the risk of liver damage and hyperlipidemia in patients with CKD."
Major concerns:
The conclusion is unfortunately not well-founded and is not based on any data related to CKD. The authors did not onvestigate CKD at all and did not investigate any disease model. The only animal experiments performed were the administration of cyanate to the healthy animals. Therefore, the conclusion is misleading.
A recent study demonstrate (29113807) cyanate administration improves insulin sensitivity and hepatic function. The authors ignored the study and need to discuss why their results contradic the study and what are the limitations of their study.
What was the rational of using 70 mice for the study? Are the data presented in Figs 1 and 2 are from 70 mice? If not please state the exact number of animals in each data set.
What was the daily intake of the drug by each animal? and did authors measure the plasma levels of the drug? Do these levels relate to human pathologies?
Fig2. How and when these tests were performed? The figure legend states three experiments, does this means from three livers? If yes, the "n" is too low to perform a statistical test.
Please state which statistical test were performed for each set of experiments.
Fig4. From the representative blots it is hardly to conclude a ~70% increase in pS6K. Please correct the labeling in D. Also state how many animals were used for these data?
Fig.5B Please provide higher magnification.
Fig.5C It is hard to conclude any change.
The "n" is missing in every figure.
Author Response
Dear Reviewers:
Thank you for your letter and for the reviewers’ comments concerning our manuscript entitled “Cyanate induce oxidative stress injury and abnormal lipid metabolism in liver through Nrf2/ HO-1” (ID: molecules-580351). Those comments are all valuable and very helpful for revising and improving our paper, as well as the important guiding significance to our researches. We have studied comments carefully and have made correction which we hope meet with approval. Revised portion are marked in red in the paper. The main corrections in the paper and the responds to the reviewer’s comments are as flowing:
Point 1: The conclusion is unfortunately not well-founded and is not based on any data related to CKD. The authors did not investigate CKD at all and did not investigate any disease model. The only animal experiments performed were the administration of cyanate to the healthy animals. Therefore, the conclusion is misleading.
Response 1: CKD is characterized by the progressive retention of metabolites, many of which have adverse effects on numerous organs. Urea is a kind of uremic toxins and may increase up to 100 mmol/l in patients with renal failure (24940796), which palys very important pathological role in atherosclerosis(25682038), depression (21750947), insulin resistance (19955654) of CKD patients by generation of ROS. Cyanate is converted from 0.8% of urea and induced atherosclerosis by carbamoylation preteins or effect eNOS to promote endothelial dysfunction in CKD (21750947). So it’s meaningful to detect the pathological role and mechanism of cyanate in different organs of CKD patient. However, a recent study demonstrates (29113807) cyanate administration improves insulin sensitivity and hepatic function, which is contradictory with other studies. So we did not just investigate cyanate level in CKD patients and directed use animal model to detect the role of cyanate to liver.
Point 2: A recent study demonstrates (29113807) cyanate administration improves insulin sensitivity and hepatic function. The authors ignored the study and need to discuss why their results contradict the study and what are the limitations of their study.
Response 2: We had mentioned this document in the previous manuscript, but did not specifically discuss it in detail. Because we used different concentrations of cyanate in test, in their study, mice were treated with cyanate (1 mg/mL in drinking water), while we used 10 mg/ml cyanate in drinking water to feed the mice. Very interesting, we both found the body weight decreased in mice fed with cyanate water. Their experimental results show that cyanate has no effect on the nervous system of mice to eliminate weight loss by reduced eating and our study just focused on the effects of cyanate on liver. The differences of experimental results between us maybe the concentration of cyanate. Our study use high concentration of cyanate to mimic the plasma level in CKD patients, and the concentration of cyanate in their study was far lower than expected LD50 (1500 mg/kg) for oral intake. The different results between high and lower concentration of cyanate indicated it maybe double-edged sword to liver in vivo. What is the banlance point of advantage and disadvantage of cyanate to liver need further researches to confirm.
Point 3: What was the rationale of using 70 mice for the study? Are the data presented in Figs 1 and 2 are from 70 mice? If not please state the exact number of animals in each data set.
Response 3: We used 70 mice throughout the study because samples were used for morphological observation (slice staining), electron microscope, enzyme measurements, and biomolecular tests. In order to avoid individual differences as much as possible, experiments require biological and technical repeatability. In our study, the mice were randomly assigned to two groups, 40 for the cyanate group and 30 for the control group. The data presented in Figure 1 was from 70 mice. As the differnrent requirement of sample preparation for morphological staining, electron microscope, enzyme activity test and molecular test, we subdivide cyanate-treated mice and normal mice into 4 subgoup, each subgroup contained at least 7 or 8 mice. The number of animal experiments was shown in the revised article.
Point 4: What was the daily intake of the drug by each animal? and did authors measure the plasma levels of the drug? Do these levels relate to human pathologies?
Response 4: Thanks so much for your meaningful comments. The mice were given 1% cyanate drinking water (ie, a 10 mg/ml cyanate concentration). Due to unstable characteristic and very low plasma concentration of cyanate which quantified by liquid chromatography-mass spectrometry (LC–MS/MS), the daily intake of mice and the plasma drug concentration were not measured in this study, which is a defect of our experiment. Previous studies (8598126) have shown that urea levels can exceed 100mM in patients with chronic renal failure, and cyanate is formed via the decomposition of urea. Further studies (24940796),( 22462773) indicated that myeloperoxidase-catalyzed oxidation of thiocyanate and myeloperoxidase-induced accelerated decomposition of urea may generate locally high cyanate levels in patient with CKD. In addition, the study (17828273) showed that patients with CKD were themselves in a low-inflammation state, so that a higher concentration of cyanate was locally present in patients with CKD. Furthermore, the study (24940796) exposed mice with 5 mg/ml sodium cyanate aerosol particles by inhalation daily 2.5 h for a period of 3 weeks and induced fourfold increase in plasma HCit levels reaching levels previously observed in patients with CKD. Inhalation is a more precise and efficient modelling method, we estimate the biotransformation and loss of cyanate water and use 10 mg/ml cyanate concentration.
Point 5: Fig2. How and when these tests were performed? The figure legend states three experiments, does this means from three livers? If yes, the "n" is too low to perform a statistical test.
Response 5: After the mice were sacrificed, the liver was perfused with normal saline (0.9% NaCl, containing 0.16 mg/ml sodium heparin) to remove the blood samples, and tissue samples were obtained (n≧30 per group). Take appropriate amount of tissue samples, add pre-cooled PBS and homogenize in an ice bath, then centrifuge the homogenate at 4 ° C, take the supernatant as the sample to be tested, measure the protein concentration of the supernatant with BCA kit, according to the instructions The procedure was carried out to determine the activities of SOD, CAT and the contents of MDA and NO. Totally 10 livers from ayanate group and 8 livers from control group were performed enzyme measurements. Each sample was repeated three times for detection. We have added this section to the revised article.
Point 6: Please state which statistical test were performed for each set of experiments.
Response 6: Thanks for your valued suggestion and we explained in the statistical method. in this study, One-way ANOVA was used for comparison of mean values across the groups and multiple comparisons were made by LSD test.
Point 7: Fig4. From the representative blots it is hardly to conclude a ~70% increase in pS6K. Please correct the labeling in D. Also state how many animals were used for these data?
Response 7: Your opinion is very important! We have repeated the WB test on the expression level of pS6K and S6K in livers of mice, renewed statistical results as “51% increase in pS6K” and replaced band in the revised article. The number of animal in experiments we have also stated in the revised article.
Point 8: Fig.5B Please provide higher magnification.
Response 8: Well, thank you for your suggestion. We have corrected in the revised article.
Point 9: Fig.5C It is hard to conclude any change.
Response 9: Thank you for your precious suggestion. we applied a higher magnification image to see the differences in the revised article. It can be seen in Figure 5C that the lipid droplets in cyanate-loaded cells were stained red by the oil red O, and the lipid droplets were significantly reduced in the cells treated with TBHQ.
Point 10: The "n" is missing in every figure.
Response 10: Thanks for your valuable suggestion. We have already added it in the revised article.
We tried our best to improve the manuscript and made some changes in the manuscript. These changes will not influence the content and framework of the paper. And here we did not list the changes but marked in red in the revised paper.
We appreciate for Editors/Reviewers’ warm work earnestly and hope that the correction will meet with approval.
Once again, thank you very much for your comments and suggestions.
Best regards!
Authors: Jian-hua Ran, Ling Hu.
Reviewer 2 Report
Using a chronic exposure model, the authors examined the effects of cyanate to cause oxidative stress, abnormal lipid metabolism, and the effects on antioxidant enzymes including Nrf2 signaling, and on metabolic signaling pathways including mTOR and AMPK. They made several interesting and novel findings: cyanate caused abnormal lipid metabolism, decreased Nrf2 expression, and increased mTOR signaling. They also showed that DMNQ could prevent some of above effects caused by cyanate in cell model.
Major concerns
1 Based on the evidence that Nrf2 protein and HO-1 in liver was decreased upon cyanate exposure, they drew the conclusion that the cyanate caused oxidative stress and abnormal lipid metabolism through decreasing Nrf2/HO1 pathway. However, this statement is based on deduction and lacks direct evidence, and current evidence only suggests the potential involvement of Nrf2/HO1 signaling. Although there was some evidence from cell model by using DMNQ, evidence is not solid. For a clear conclusion, more evidence is needed, such as whether the effects of cyanate could be rescued by Nrf2 overexpression or exacerbated by Nrf2 knockout. Otherwise the conclusion should be modified.
2 The total Nrf2 protein was decreased in Fig 4A, how about the Nrf2 activity? Theoretically, Nrf2 is activated and nuclear Nrf2 should be increased in response to oxidative stress, which occurred upon cyanate exposure. Is the nuclear Nrf2 and Nrf2-ARE binding activity decreased or increased by cyanate?. discuss the potential mechanism why Nrf2/HO-1 was down regulated in mice exposed to cyanate.
3 Line 326, nuclear Nrf2 was significantly decreased? Nrf2 looks increased by cyanate in Figure 7. A densitometry graph should be used to summarize Nrf2 change.
4 Figure 1, ALT, AST, and ALP are well-recognized marker of liver damage, why their levels were decreased when live injury was observed? This should be explained in the discussion.
Minor concerns
Line 55, change “demonstrated” to “demonstrate)
Line 57, change “ the current research has” to “these previous studies have”
Line 60, “Cyanate is a uremic toxin converting by 0.8% of the molar concentration of urea” “Cyanate is a uremic toxin that is converted from 0.8% of the urea”
Line 58, start a new paragraph from “CKD is characterized…”
Line 67, “…amino acid and protein modifications by affecting… ” to “…amino acids and proteins modifications that affects…”
Line 75, “cyanate in patients with CKD that can…” to “whether cyanate in patients with CKD can…”
Line 84 “and a phase 2 detoxifying enzyme” to “and phase 2 detoxifying enzymes”
Line 88, “Generally” to Normally”
Line 92, “encoding and phase 2 93 detoxifying molecules to exert an antioxidant effect” to “encoding antioxidant and phase 2 detoxifying enzymes”
Line 115, “can significantly reduce…” to “significantly reduced”
Line 141, line 162, and other legends, instead of using “three separate experiments”, indicating N value or from how many mice was the data collected.
Line 147- line 153, show the full name instead of the abbreviation if it is used for the first time in the paper.
Figure 2 legend, briefly describe the detection method for SOD, CAT, MDA and NO.
Figure 4D, there were two pS6K/S6K in the figure
Line 175, not clear, should it be cyanate increased ROS levels and caused lipid…?
Line 176, “..by inhibiting the Nrf2 pathway in vivo”, there is no direct evidence to support this statement. Above evidence only suggests a potential association.
Line 177, “The mechanism of the effects of cyanate on liver needs to explored in vitro”, “we explored the potential mechanism of how cyanate caused liver damage in vitro”.
Line 184, TBHQ is not a specific agonist for Nrf2, it is a recycling quinone that generates superoxide and H2O2 in cells
Figure 5, also show the DCF data with a bar graph; pretreated with DMNQ for how long before being exposed to cyanate?
Line 210, was
Figure 6, describe how long the cells were pretreated with DMNQ before cyanate exposure
Line221-222, it is a repeat of line 184
Figure 7, does it show that the fluorescence signal of Nrf2 in the cytoplasm is increased by cyanate, DMNQ, and also cyanate+DMNQ? And Nrf2 nuclear translocation loos decreased by cyanate? Using a bar graph to show the desitometry data.
Line 250, is High-density lipoprotein (HDL) decreased or increased?
Line 257, repeated “progression”
Line263-264, does it mean cyanate decreased liver damage? This data is contrast to the findings of hepatic lobular structural disorders, hepatic steatosis and lipid deposition.
Line 241-260, most of it is repeating the introduction, make it short
Line 286-294, has been described in line 124-132.
Line 298, try to explain why SOD and catalase activity was decreased by cyanate
Line 308, detoxification enzymes
Line 309, delete antioxidant response element
Line 314, delete “by”
Line 315, delete “antioxidant response element”, use ARE
Line333-334, it is irrelevant to this study. Delete it
Line 350, mammalian target of rapamycin
Line 357, DMNQ is not a inhibitor of Nrf2. As a pro-oxidant, DMNQ generates ROS in cells and actually activates Nrf2, leading to enhanced antioxidant defense.
Line 402, “and is free”?
Line 426, “thrice”?
Line 427, “(Danvers, 3 Trask Lane, USA.)”, in the form of (city, state, country). Correct others.
Line 458, describe the sex of mice
Author Response
Dear Reviewers:
Thank you for your letter and for the reviewers’ comments concerning our manuscript entitled “Cyanate induce oxidative stress injury and abnormal lipid metabolism in liver through Nrf2/ HO-1” (ID: molecules-580351). Those comments are all valuable and very helpful for revising and improving our paper, as well as the important guiding significance to our researches. We have studied comments carefully and have made correction which we hope meet with approval. Revised portion is marked in red in the paper. The main corrections in the paper and the response to the reviewer’s comments are as flowing:
Point 1: Based on the evidence that Nrf2 protein and HO-1 in liver was decreased upon cyanate exposure, they drew the conclusion that the cyanate caused oxidative stress and abnormal lipid metabolism through decreasing Nrf2/HO1 pathway. However, this statement is based on deduction and lacks direct evidence, and current evidence only suggests the potential involvement of Nrf2/HO1 signaling. Although there was some evidence from cell model by using DMNQ, evidence is not solid. For a clear conclusion, more evidence is needed, such as whether the effects of cyanate could be rescued by Nrf2 overexpression or exacerbated by Nrf2 knockout. Otherwise the conclusion should be modified
Response 1: Thank you very much for your valuable suggestions, we have modified our conclusions according to your advice.
Point 2: The total Nrf2 protein was decreased in Fig 4A, how about the Nrf2 activity? Theoretically, Nrf2 is activated and nuclear Nrf2 should be increased in response to oxidative stress, which occurred upon cyanate exposure. Is the nuclear Nrf2 and Nrf2-ARE binding activity decreased or increased by cyanate?. discuss the potential mechanism why Nrf2/HO-1 was down regulated in mice exposed to cyanate.
Response 2: Theoretically, studies(29255090) have shown that under normal conditions, Nrf2 binds to Keap1 in the cytosol, and when exposed to electrophiles or reactive oxygen species, the cysteine residue of Keap1 is modified and altered. The changes of conformation cause Nrf2 to be released, enters the nucleus, and binds to ARE to promote expression of the corresponding target gene. However, it was later found(29913224) that the inducer did not dissociate Keap1-Nrf2, and that Nrf2 could be degraded by ubiquitination, resulting in a decrease in the level of Nrf2. In our experiments, western blot results showed that the levels of Nrf2 in the liver tissue and cells of the cyanate group decreased, the results of cellular immunofluorescence showed that nuclear Nrf2 was significantly reduced in cyanate-loaded cells, indicating that cyanate may also ubiquitinate Nrf2 to cause degradation while stimulating the production of reactive oxygen species. The reduction of nuclear Nrf2 translocation weakened the antioxidant capacity of the cells and aggravatedthe oxidative stress.
Point 3: Line 326, nuclear Nrf2 was significantly decreased? Nrf2 looks increased by cyanate in Figure 7. A densitometry graph should be used to summarize Nrf2 change.
Response 3: Thank you very much for your valuable suggestions! We are very sorry for our negligence of the brightness of the image, we have changed it in the manuscript. In Fig. 7 E, we can clearly see that there is no green fluorescence in the nucleus of the cyanate-loaded cells , indicating that Nrf2 does not enter the nucleus to exert its activity. Statistical charts of fluorescence density analysis have also been added to the article.
Point 4: Figure 1, ALT, AST, and ALP are well-recognized marker of liver damage, why their levels were decreased when live injury was observed? This should be explained in the discussion.
Response 4: Thank you very much for your precious comments! We confirmed our biochemical analysis report and statistics and found that the two sets of data were reversed when drawing. We are very sorry for our negligence, we have made corrections in the article. Thank you for pointing out our question, which is very important for us to modify the article!
Point 5: Minor concerns.
Response 5: Thank you very much for reading our article carefully and making valuable suggestions, and these concerns have been corrected in the article.
We tried our best to improve the manuscript and made some changes in the manuscript. These changes will not influence the content and framework of the paper. And here we did not list the changes but marked in red in revised paper. And the revised article is in the attachment.
We appreciate for Editors/Reviewers’ warm work earnestly, and hope that the correction will meet with approval.
Once again, thank you very much for your comments and suggestions.
Best regards!
Authors: Jian-hua Ran, Ling Hu.
Round 2
Reviewer 2 Report
Comments
The manuscript is improved. Scientifically, the statements and conclusion are supported by the data. However, more improvements are needed before being accepted for publication.
Concerns
Paragraph structure. Some paragraphs are not well organized; some sentences are repeat description, some are irrelevant to the topic, or lack of logic.
Grammar errors.
Line 121, controversial to Figure 1 which showed activities of ALT and other biomarkers of live injury were increased by cyanate.
Line 136-143, correct the grammar errors and move the paragraph to the Method
Line 160 and other legends, change to “n≥8 per group”, delete" of three separate experiments”.
Line 185, inhibited
Line 186, translocated into
Line 194, change the title of legend of figure 4, “expression of Nrf2 pathway” is inappropriate as a conclusive statement of the figure
Line 201, reduced the activity of,
Line 205, briefly describe how CCK-8 assay measures cell viability
Line 287, repeated progression
Line 302, decrease of body weight was not inferred, it was observed.
Line 331-332, delete “SOD… in the body”. It is a repeat.
Line334, indicates
Line 370, was
Line 371, add, the exact mechanism of how cyanate decrease Nrf2 expression remains to be determined in the future.
Line 414, the results suggest
Line 490-496, grammar